# Effect of Organic and Synthetic Fertilizers on Nitrate, Nitrite, and Vitamin C Levels in Leafy Vegetables and Herbs

**DOI:** 10.3390/plants14060917

**Published:** 2025-03-14

**Authors:** Nga Thi Thu Nguyen, Bac Xuan Nguyen, Nasratullah Habibi, Maryam Dabirimirhosseinloo, Leonardo de Almeida Oliveira, Naoki Terada, Atsushi Sanada, Atsushi Kamata, Kaihei Koshio

**Affiliations:** 1Faculty of International Agriculture and Food Studies, Tokyo University of Agriculture, Sakuragaoka 1-1-1, Setagaya-ku, Tokyo 156-8502, Japan; nasratullah.habibi14@gmail.com (N.H.); 13423403@nodai.ac.jp (M.D.); leonardoaaloliveira@gmail.com (L.d.A.O.); nt20436@nodai.ac.jp (N.T.); a3sanada@nodai.ac.jp (A.S.); koshio@nodai.ac.jp (K.K.); 2Faculty of Food Science and Technology, Vietnam National University of Agriculture, Trau Quy, Gia Lam, Hanoi 12406, Vietnam; nxbaccntp@vnua.edu.vn; 3Faculty of Agriculture, Balkh University, Balkh 1701, Afghanistan; 4Faculty of Agriculture, Tokyo University of Agriculture, Isehara Farm, 1499-1 Maehata, Sannnomiya, Isehara 259-1103, Japan; ak207913@nodai.ac.jp

**Keywords:** health risk, leafy vegetables, nitrate, nitrite, nitrogen accumulation, organic fertilizer, synthetic fertilizer, vitamin C

## Abstract

This study investigated the accumulation of nitrate and nitrite, as well as the vitamin C content, in various leafy vegetables, including amaranth greens, katuk, morning glory, squash leaves, vine spinach, leaf lettuce, romaine lettuce, Vietnamese basil, Vietnamese perilla, komatsuna, leeks, and spinach, grown with either organic or synthetic fertilizers. The findings indicate that the type of fertilizer significantly influences nitrate accumulation and vitamin C content in these vegetables. Organic fertilizers were found to reduce nitrate levels and increase vitamin C content in amaranth greens, katuk, morning glory, squash leaves, vine spinach, leaf lettuce, Vietnamese basil, Vietnamese perilla, and spinach compared to the results for synthetic fertilizers. However, high nitrate concentrations in leaf lettuce, komatsuna, and spinach pose potential health risks. The study also identified elevated nitrate levels in vegetables that are not currently regulated. Furthermore, more than half of the samples contained nitrite, for which no maximum permissible level has been established. These findings underscore the importance of organic vegetable cultivation in enhancing both human health and environmental sustainability.

## 1. Introduction

Vegetables are a crucial source of essential nutrients such as vitamins [1,2], minerals, carbohydrates, and beneficial compounds like polyphenols, flavonoids, and glucosinolates [3,4,5]. However, they also naturally contain nitrate (NO_3_), which is absorbed from the soil and converted into essential proteins for plant growth. The potential health hazards of NO_3_ and nitrite (NO_2_) are well-studied. While NO_3_ is generally considered less toxic than NO_2_, about 5% of dietary NO_3_ is converted to NO_2_ in humans [6,7], which can reduce the oxygen supply to the body. Elevated NO_2_ levels can cause methemoglobinemia (blue baby syndrome) [8]. Furthermore, NO_2_ can react with amines and amides to form nitrosamines and nitrosamides—potentially carcinogenic N-nitroso compounds (NOCs) [9,10]. Recently, research has explored the potential positive role of inorganic NO_3_ and NO_2_ in cardiovascular disease [11]. Consequently, the NO_3_ content in vegetables has become a concern due to its potential health implications and relationship to agricultural practices.

The NO_3_ level in vegetables is influenced by genetic, environmental, and agricultural factors. Leafy vegetables, such as spinach and lettuce, generally contain higher NO_3_ levels [12]. Among these factors, nitrogen fertilization has been identified as a key determinant of NO_3_ levels in vegetables [13]. Excessive nitrogen (N) fertilization leads to higher NO_3_ content in vegetables, regardless of the type of fertilizer used. Additionally, overfertilization increases the risk of groundwater contamination with NO_3_ [14].

Recent trends indicate a growing preference for organic fertilizers due to concerns about the health and environmental impacts of synthetic fertilizers. In theory, organic vegetables should contain lower NO_3_ levels than those grown using conventional methods [15,16,17,18]; however, De Martin and Restani (2003) found that organically grown green salad and rocket contained significantly higher NO_3_ levels than conventionally produced types [19]. Similarly, Guadagni et al. (2005) reported higher NO_3_ levels in organic watercress compared to those in conventionally cultivated samples [20]. Additionally, some studies have shown that organically grown crops may contain NO_3_ and NO_2_ levels similar to those of their conventionally grown counterparts [21,22,23]. Despite these findings, there is limited evidence regarding whether organic farming consistently results in lower NO_3_ or NO_2_ accumulation compared to that resulting from conventional methods. Further research is needed to clarify the effects of different fertilizers on the nutrient and toxin profiles of vegetables.

Although much research has focused on NO_3_ concentrations due to their role as precursors to NO_2_, existing regulations, such as European Union (EU) Regulation 1258/2011, primarily set NO_3_ limits only for spinach, lettuce, and rucola [24]. This regulation does not extend to other widely consumed Asian vegetables, including amaranth greens, katuk, morning glory, and vine spinach. Moreover, NO_2_ levels, which pose significant health risks, remain unregulated in vegetables. Additionally, the formation of harmful NOCs is influenced by various dietary elements, including vitamin C. Some studies suggest that consuming vitamin C-rich vegetables alongside NO_3_-containing foods can facilitate the conversion of NO_2_ into nitric oxide (NO), thereby inhibiting the formation of harmful N-nitroso compounds [25].

The present study aims to assess NO_3_, NO_2_, and vitamin C content in vegetables grown locally in Vietnam and Japan. The objectives of this study are (1) to examine the impact of organic versus synthetic fertilizers on nitrogen accumulation and vitamin C content in leafy vegetables commonly consumed in South and Southeast Asia and (2) to evaluate the potential health risks associated with consuming these vegetables.

To our knowledge, this is the first study to investigate whether significant differences in NO_3_, NO_2_, and vitamin C levels exist between leafy vegetables and herbs grown using organic versus synthetic fertilizers in South and Southeast Asia. These findings will contribute to a better understanding of how different fertilization practices affect nutrient and toxin levels in commonly consumed leafy vegetables and herbs and will provide valuable insights for consumer health risk assessments.

## 2. Results

### 2.1. Vegetables and Herbs in Northern Vietnam

The NO_3_ accumulation (mg/kg fresh weight—FW) in observed vegetables and herbs from Northern Vietnam from July 2023 to August 2024 is presented in Figure 1 and Table 1.

As shown in Figure 1, vegetables grown with synthetic fertilizer generally showed a higher NO_3_ content than that of their organic counterparts, except for Vietnamese basil and Vietnamese perilla. The results showed a significant variation in the average NO_3_ levels between the two fertilizer types and different vegetables. The highest NO_3_ level was found not only in leaf lettuce (synthetic fertilizer: 4232 ± 90 mg/kg FW; organic fertilizer: 3654 ± 292 mg/kg FW) but also in amaranth greens (synthetic fertilizer: 4099 ± 698 mg/kg FW), and there was no significant difference between them (*p* > 0.05). Amaranth greens, katuk, and vine spinach did not accumulate NO_3_ when grown with organic fertilizer.

The NO_3_ level under organic fertilizer was lower in leaf lettuce, but the difference was not statistically significant compared to that of the synthetic fertilizer (*p* = 0.26). In leafy vegetables, including amaranth greens, katuk, morning glory, squash leaves, vine spinach, and romaine lettuce, the organic fertilizer treatment resulted in lower NO_3_ levels than those from the synthetic type, whereas in herbs like Vietnamese basil and Vietnamese perilla, the trend was reversed (*p* < 0.05). This may be due to the inherent differences between leafy vegetables and herbs [18]. Although these findings indicate that fertilizer type influences NO_3_ accumulation in leafy vegetables and herbs, organic fertilizer generally led to lower NO_3_ content in leafy vegetables, particularly in amaranth greens (Table 1). This can be explained by the fact that NO_3_ from synthetic fertilizers is readily available and quickly absorbed, often in excess, leading to higher NO_3_ levels in leafy vegetables.

The NO_2_ content of these vegetables is presented in Figure 2 and Table 2.

As can be seen in Figure 2, vine spinach, leaf lettuce, romaine lettuce (both synthetic and organic), and squash leaves (synthetic) did not contain NO_2_. Katuk grown with organic fertilizer had the highest accumulation of NO_2_, with 79 ± 7 mg/kg FW. The second highest accumulation was for amaranth greens (42 ± 2 mg/kg FW NO_2_) and Vietnamese perilla (45 ± 3 mg/kg FW NO_2_) grown using organic fertilizers.

The average levels of NO_2_, except in amaranth greens, katuk, Vietnamese basil, and Vietnamese perilla, were very similar (*p* > 0.05) for both the organic and synthetic fertilizers (Table 2). It seems that only the NO_3_ levels were significantly affected by the type of fertilizer. This means that organic fertilizer increases the NO_2_ content in amaranth greens, katuk, Vietnamese basil, and Vietnamese perilla.

Vitamin C is the most elevated element in Vietnamese vegetables and herbs, up to 1600 mg/100 g fresh weight (Figure 3). The highest vitamin C value is 1601 ± 66 mg/100 g FW in katuk (organic), followed by 1165 ± 55 mg/100 g FW in amaranth greens (organic) and 1149 ± 63 mg/100 g FW in katuk (synthetic). The next highest values were found in amaranth greens (synthetic) (959 ± 99 mg/100 g FW), Vietnamese perilla (organic) (844 ± 17 mg/100 g FW), and Vietnamese perilla (synthetic) (802 ± 11 mg/100 g FW). The lowest vitamin C value was found in romaine lettuce (synthetic) (12 ± 0 mg/100 g FW) and romaine lettuce (organic) (14 ± 1 mg/100 g FW). The remainder displayed vitamin C contents from 73 ± 5 mg/100 g FW to 605 ± 58 mg/100 g FW (Table 3).

Vegetables grown using synthetic fertilizer exhibited lower vitamin C levels than their organic counterparts, and they were significantly different in all types of leafy vegetables and herbs grown in our study (*p* < 0.05). Previous research has shown that when nitrogen availability in the soil is restricted, levels of vitamin C (ascorbic acid) are likely to rise [26]. Our study results validate this proposal, as we found higher levels of vitamin C in leafy greens and herbs that were grown with organic fertilizers.

### 2.2. Vegetables in Kanto Region of Japan

The NO_3_ accumulation in the observed vegetables from September 2023 to July 2024 is presented in Figure 4, Table 4 and Table 5. None of the vegetables contained NO_2_.

A similar trend in NO_3_ content was observed between vegetables in Japan and Vietnam, where vegetables grown with synthetic fertilizers had higher NO_3_ levels than those grown with organic fertilizers (Figure 4). Organic fertilizer resulted in lower NO_3_ levels in komatsuna blades, leek stems, spinach blades, and spinach midribs and petioles (*p* < 0.005) (Table 4). There was also a significant difference in NO_3_ levels between different parts of these vegetables (*p* < 0.005) (Table 5). NO_3_ tends to accumulate more in the midrib and petiole than in the blade of komatsuna and spinach and more in the stem than in the leaf of leeks. This can be explained by their structural and functional roles. The midrib and petiole provide support and transport nutrients and water, which may contribute to their higher NO_3_ content, as they serve as conduits and storage areas. The leaf blade, on the other hand, is the primary site for photosynthesis and other metabolic activities, meaning that it contains higher levels of enzymes involved in NO_3_ reduction and assimilation, which convert NO_3_ into other compounds, resulting in lower NO_3_ levels in the blade.

The vitamin C content in these vegetables is presented in Figure 5, Table 6 and Table 7.

The vitamin C content in spinach grown with organic fertilizer is the highest (273 ± 15 mg/100 g) (Figure 5) and was significantly different from that of plants grown with synthetic fertilizer (*p* < 0.001). Normally, the vitamin C value of plants grown using organic fertilizer is always higher than that for plants grown using synthetic types; however, it was not enough to make it significantly different (*p* > 0.5) in komatsuna (Table 6). Therefore, organic fertilizer only efficiently increased vitamin C content in leeks and spinach.

### 2.3. Daily Intake of Nitrogen Compound and Vitamin C of Studied Vegetables

The daily intake of NO_3_, NO_2_, and vitamin C for the studied vegetables is presented in Figure 6, Figure 7 and Figure 8.

The European Commission’s Scientific Committee on Food (SCF) and the Joint Expert Committee of Food and Agriculture (JECFA) have set acceptable daily intakes (ADI) of 0–3.7 mg/kg body weight (BW) for NO_3_ ion and 0–0.06 mg/kg BW for NO_2_ ion for the human body [27]. For a 60 kg adult, these values are equivalent to 222 mg of NO_3_ and 3.6 mg of NO_2_ per day.

Our investigation indicated that the daily intake of NO_3_ in eight leafy vegetables (katuk, morning glory, squash leaves, vine spinach, romaine lettuce, Vietnamese basil, Vietnamese perilla, and leeks) were negligible as compared to tolerable daily intake standard set by the SCF and JECFA (Figure 6). Hence, it can be concluded that there is no risk in consuming these leafy vegetables. Only amaranth greens (synthetic fertilizer), leaf lettuce, and komatsuna samples contained higher concentrations of NO_3_, which were about 1.8-fold, from 1.6- to 1.9-fold and from 1.7- to 3.4-fold higher than the ADI, respectively.

Figure 7 indicated that the NO_2_ daily intake was significantly higher than the tolerable daily intake standard set by the SCF and JECFA in amaranth greens (organic), katuk (synthetic), and Vietnamese perilla (organic) [24]. The daily intake of NO_2_ for these leafy vegetables was about 1.1- to 1.3-fold higher than the safe limits value of NO_2_ in humans. Hence, it can be concluded that there would be a risk when consuming these vegetables.

Figure 8 also shows that the recommended daily intake of vitamin C for breastfeeding women is 120 mg per day [28] when consuming 100 g of amaranth greens, katuk, morning glory, squash leaves, Vietnamese basil, Vietnamese perilla (both synthetic and organic), vine spinach (organic), and leaf lettuce (organic). The same quantity of the other vegetables will not provide enough vitamin C for this population and they need to provide other sources of vitamin C to get enough requirement for daily intake.

## 3. Discussion

This study explored a wide range of leafy vegetables, such as amaranth, katuk, komatsuna, lettuce, leeks, morning glory, vine spinach, and herbs, some of which are not regulated for NO_3_ levels. These vegetables are widely consumed in South and Southeast Asia, commonly used fresh in salads, spring rolls, sandwiches, and noodle soups like pho, as well as in boiled, stir-fried, or hot pot dishes.

This is the first report to compare the NO_3_/NO_2_ and vitamin C levels of a wide range of leafy vegetables and herbs from Japan and Vietnam under the effects of organic versus synthetic fertilizers.

### 3.1. Vegetables in the Northern Vietnam

The vegetables selected for this study are among the most consumed and were available during the collection period. In Northern Vietnam, these vegetables fall into two categories. The first category includes amaranth greens, katuk, morning glory, squash leaves, and vine spinach, typically cooked before consumption. The second category consists of leaf lettuce, romaine lettuce, Vietnamese basil, and Vietnamese perilla, commonly eaten fresh as salads.

#### 3.1.1. Nitrate Content

The levels of NO_3_ present in amaranth greens, for which no established maximum exists, are noteworthy findings that require discussion. The results displayed a NO_3_ concentration similar to that of lettuce when subjected to synthetic fertilizers. Both types were slightly above the acceptable limit set by EU regulations (Regulation (EU) No 1258/2011), which is 4000 mg/kg FW, for lettuce harvested from April 1 to September 30 [24]. Amaranth greens are typically eaten after cooking, meaning that food processing methods like washing and cooking can influence the NO_3_ concentrations in vegetables. Since NO_3_ dissolves in water, washing and soaking vegetables can lead to a reduction of around 10 to 15% in NO_3_ content. Similarly, boiling can also decrease NO_3_ levels, with reductions that range from 16 to 79% [27]. Consequently, this issue may not be significant for amaranth greens. However, the impact of heat treatment on lowering NO_3_ does not apply to leaf lettuce, leaving only the washing process to affect these levels. This highlights another crucial reason why agrotechnical measures should be followed during cultivation, as they can help achieve lower NO_3_ levels in vegetables.

The NO_3_ concentration found in the romaine lettuce was in the same range as those determined by the European Food Safety Authority (EFSA), i.e., from 167 to 2200 mg/kg [27]. The levels for the remaining vegetables were significantly below the recommended NO_3_ levels; thus, from the point of view of NO_3_, katuk, morning glory, squash leaves, vine spinach, romaine lettuce, Vietnamese basil, and Vietnamese perilla are safe when grown using both types of fertilizer.

#### 3.1.2. Nitrite Content

As expected, the average levels of NO_2_ were lower when compared with those of NO_3_. Our findings were similar to those of Alfredo et al. [21], who reported a significantly higher NO_2_ content in organic than in conventionally grown red lettuce [21].

Our results show that NO_2_ was detected in 61% of the samples we examined in our research. It could be quite astonishing to find NO_2_ in certain leafy vegetables and herbs exceeding 40 mg/kg of fresh weight. These figures are considerably higher than the typical level found in fresh leafy vegetables, which is around 2 mg/kg [29]. Research conducted by Menard et al. (2008) indicated that the highest NO_2_ content in lettuce was 25 mg/kg, while for spinach, it was 220 mg/kg [30]. The elevated concentrations might result from microbial processes converting NO_3_ to NO_2_ at ambient temperatures, or even under refrigerated conditions. In Vietnam, many supermarkets display fresh produce, such as amaranth, katuk, morning glory, Vietnamese basil, and Vietnamese perilla, at room temperature during the day. This environment may facilitate the transformation of NO_3_ into NO_2_, leading to NO_2_ concentrations that exceed typical findings in the existing literature, particularly for those plant types that naturally accumulate high NO_3_ levels.

Leaf lettuce and romaine lettuce were intended for fresh consumption, and they were free of NO_2_, so they are considered safe for consumption. Vietnamese basil and Vietnamese perilla are herbs that are eaten raw; thus, only handling and storage would impact NO_2_ levels. Overall, the losses of NO_2_ were greater than for NO_3_ when applying different preliminary processing and heating methods [27], and the consumed amount for herbs is much smaller than that for leafy vegetables, so these levels may not be problematic for these herbs.

#### 3.1.3. Vitamin C Content

Research indicates that various substances, such as vitamin C, can diminish the creation of NOCs by obstructing the nitrosation process. This obstruction takes place under the stomach’s acidic conditions when both NO_2_ and amines are present. Vitamin C converts HNO_2_ into NO, which does not contribute to nitrosation. Additionally, it interacts more rapidly with N_2_O_3_ that with amines, thereby reducing the production of NOCs. A dosage of 1 g of ascorbic acid fully stopped the rise in N-nitrosoproline excretion in urine, which serves as a marker for the internal generation of NOCs [25]. With a significant vitamin C content reaching 1600 mg per 100 g fresh weight, these vegetables seem to be able to adequately inhibit the synthesis of nitrosamines.

The possible biochemical processes through which organic fertilizers could lead to increased amounts of vitamin C (ascorbic acid) and reduced NO_3_ levels in vegetables include various factors linked to how plants absorb nutrients, their metabolic activities, and how they respond to stress. Organic fertilizers generally provide N through organic matter, like compost or manure, and they release N at a slower rate compared to that of synthetic fertilizers, which supply NO_3_. This slow N release from organic fertilizers leads to reduced NO_3_ buildup in plant tissues [18]. The production of vitamin C and the uptake of NO_3_ occur through different pathways and metabolic functions in the plant. When exposed to high NO_3_ concentrations, plants often focus on converting available resources into biomass, which may limit the energy and resources devoted to producing vitamin C. Thus, plants with high levels of NO_3_ usually show reduced vitamin C content [31]. Additionally, when NO_3_ levels become excessively high, this can cause oxidative stress. In reaction to this stress, plants might redirect energy to turn NO_3_ into less harmful substances, instead of producing antioxidants like vitamin C. As a result, when plants are faced with high NO_3_ levels, their vitamin C concentrations often decline as the plant’s metabolism shifts to addressing oxidative stress and excess NO_3_.

### 3.2. Vegetables in the Kanto Region of Japan

#### 3.2.1. Nitrate Content

The NO_3_ concentrations in spinach in our study were lower than those required by EU regulations [24], and in the same range as those determined by the EFSA, i.e., from 64 to 3048 mg/kg [27].

The NO_3_ concentration found in leeks was in the same range as those determined by the EFSA, i.e., from 5 to 975 mg/kg [27].

Komatsuna is not covered by EU regulations (Regulation [EC] No. 1258/2011) [24], and with the NO_3_ value of 7467 ± 387 mg/kg FW when using synthetic fertilizer, it should be under consideration to set maximum in the regulation. The NO_3_ levels in komatsuna are clearly greater than those in lettuce and spinach, as shown in Figure 4. The same applies to amaranth greens, which are consumed after cooking. Even when reduction factor is taken into consideration (15% reduction for washing vegetables; 50% reduction for heat treatment) [27], a 100 g portion of fresh komatsuna contained an NO_3_ concentration of 261.3 mg, exceeding the ADI of 222 mg.

A similar tendency was found by Pussemier et al. (2006), who reported significant differences in the average levels of NO_3_ contents from organic and conventional produce. They reported lower levels of NO_3_ in organic (1703 mg/kg) and higher levels (2637 mg/kg) in conventional produce [32].

Also, our findings showed a NO_3_ variation between the Asteraceae (lettuce), Brassicaceae (komatsuna), Chenopodiaceae (spinach) and Amarantaceae (amaranth greens) families, which are those with the highest average levels. This result agrees with that of Santamaria (2006), who stated that families like Brassicaceae (rocket, radish, mustard, and cress), Chenopodiaceae (beetroot, Swiss card, spinach), Amarantaceae, Asteraceae (lettuce), and Apiaceae (celery, parsley) are usually the plant families (among the vegetables) with the highest NO_3_ contents. This tendency was confirmed in the present study [29].

It is commonly assumed that the NO_2_ levels in fresh leafy vegetables are usually less than 2 mg/kg FW [32]. In this study, the NO_2_ levels of vegetables from the Kanto region of Japan agreed with this assumption.

#### 3.2.2. Vitamin C Content

In contrast to NO_3_, vitamin C was accumulated in the blade and significantly higher than in the midrib and petiole (Table 7). This could be because the blade of the leaf is the primary site for photosynthesis, where the plant produces sugars and other compounds, including vitamin C. The higher metabolic activity in the blade leads to greater synthesis and accumulation of vitamin C. Vitamin C is often utilized and stored in areas of the plant where it can protect against oxidative stress and support metabolic functions. The blade, being more exposed to light and environmental stress, requires higher levels of vitamin C for protection [33].

### 3.3. Risk Assessment of the Effects of Nitrate, Nitrite, and Vitamin C in the Studied Vegetables on Human Health 

When evaluating the risk of consuming vegetables, it is essential to consider the combined effects of all compounds present, including NO_3_, NO_2_, and vitamin C, rather than assessing them separately.

According to the World Health Organization (WHO), a healthy diet includes a daily intake of 400 g of fruits and vegetables [34]; therefore, to assess the total risk for an individual weighing 60 kg consuming 200 g per day of the leafy vegetable samples or 100 g per day of the herb samples, a comprehensive table that incorporates these factors is created and presented in Table 8.

The total risk assessment indicates that the daily intake of NO_2_ significantly exceeds the tolerable daily intake standard set by the SCF and the JECFA, which is 3.6 mg of NO_2_ [27]. This excess is observed in vegetables such as komatsuna, spinach, leaf lettuce, romaine lettuce, katuk (organic), morning glory (synthetic), and squash leaves (synthetic). For a consumption of 200 g of leafy vegetables or 100 g of herbs, the total NO_2_ intake ranges from 1.2 to 6.8 times higher than the safe limit for human consumption.

Reducing the intake of katuk, morning glory, squash leaves, and romaine lettuce from 200 g to 100 g can bring the total risk below the safe limit. However, the NO_2_ levels in leaf lettuce (3.0 to 3.8 times), komatsuna (3.4 to 6.8 times), and spinach (2.6 times) remain high and cannot be mitigated by simply reducing the amount consumed. This study revealed that NO_3_ accumulates more in the midrib and petiole, while vitamin C is concentrated in the blade; therefore, a practical approach to reducing risk could involve cutting down the number of consumed vegetables and removing the midrib and petiole. This strategy could help lower NO_3_ intake, while maintaining vitamin C levels.

The ADI values for NO_3_ and NO_2_ are 3.7 mg/kg body weight/day and 0.06 mg/kg body weight/day, respectively. For a person weighing 60 kg, this translates to an ADI of 220 mg for NO_3_. With 5% of the ingested NO_3_ converted to NO_2_, this results in a NO_2_ exposure of 11 mg. The ADI for NO_2_ is 3.6 mg for a 60 kg person. Therefore, the conversion of NO_3_ to NO_2_ means that the ADI for NO_2_ is already exceeded when NO_3_ is consumed at its ADI level. This suggests that the current WHO and EU standards for NO_3_ ADI may not be sufficient to prevent all risks associated with NO_3_ exposure or to protect against the combined adverse health effects of NO_3_ and NO_2_.

## 4. Materials and Methods

### 4.1. Samples

This study investigated a range of commonly eaten vegetables in South and Southeast Asia, some of which are not covered by current NO_3_ regulations.

Between July 2023 and August 2024, nine groups of samples were collected, chosen because they were popular and available in both organic and conventional products. Sample of fresh amaranth greens, katuk, squash leaves, morning glory, vine spinach, leaf lettuce, romaine lettuce (winter), Vietnamese perilla, and Vietnamese basil were randomly acquired from supermarkets and directly from local farms, including conventional farms that use synthetic fertilizers and organic farms that use organic fertilizers derived from plant wastes in Gia Lam, Hanoi, Vietnam. Amaranth greens, katuk, morning glory, vine spinach, Vietnamese basil, and Vietnamese perilla were collected from a supermarket that displays fresh produces at room temperature during the day. Leaf lettuce and romaine lettuce were acquired from a supermarket that displays fresh produces at cool temperatures, around 18 °C. Squash leaves were provided by a local farm. At least three lettuce plants or bundles of vegetables/herbs (quantity of 200–500 g for each) of the three replicates for each vegetables/herb were analyzed as sub-samples and averaged to produce one sample data point. The location of the crops grown is an altitude of 13 m, a latitude of 21°26′38 N, and a longitude of 106°11′56 E. From July to August 2023 and 2024, the temperature range was 25–35 °C, and the humidity range was 75–90%. From February to March 2023 and 2024, the temperature range was 15–25 °C, and the humidity range was 80–90%. The weather conditions were recorded using temperature and humidity data loggers (EBI 20-T1, Ebro, Hamburg, Germany).

From September 2023 to July 2024, komatsuna, leeks, and spinach from the Ibaraki Agricultural Institute in the Kanto region of Japan were used for this study. Five vegetable plants for each type were analyzed as sub-samples and average to produce one sample data point. The location where the crops were grown is an altitude of 9 m, a latitude of 36°05′00 N, and a longitude of 140°12′00 E. From September to November 2023, the average temperature range was 11–23 °C, and the relative humidity was 70–80%; from April to July 2023 and 2024, the temperature range was 12–25 °C, and the relative humidity was 74–78%. This information was based on historical climate data for the Ibaraki Prefecture [35,36]. This part of the study aimed to assess the effects of fertilizers and the specific parts of vegetables on the accumulation of N compounds and vitamin C. In komatsuna and spinach, the blade, midrib, and petiole were evaluated, while in leeks, the leaf and stem were analyzed separately (Figure 9).

The vegetables selected for this study are some of the most consumed conventional and organic products in Vietnam and Japan between July 2023 and August 2024 (Table 9). The sampling was completed in two different seasons; however, due to variations in the types of vegetables and geographical locations, this study did not discuss the effect of season. Additionally, information on the cultivation conditions was not available.

As a rule, synthetic and organically fertilized vegetables were collected on the same day. All the commercial samples were obtained in the original package, within the shelf life of up to 7 days, as declared on the labels. After collection, the samples were cold transported to the Biochemical Laboratory at the Vietnam National University of Agriculture (VNUA) and the Tropical Horticultural Science Laboratory at Tokyo University of Agriculture (TUA). The samples were analyzed immediately to avoid elevated results due to storage.

### 4.2. Nitrate and Nitrite Analysis

The NO_3_ and NO_2_ contents in the vegetables were determined using the reflectometric method after applying a reducing agent and Griess reaction [37]. This method is based on the principle in which NO_3_ ions are reduced to NO_2_ ions by a reducing agent. The NO_2_ ions (originally present, plus reduced from NO_3_), in the presence of an acidic buffer, react with an aromatic amine to form a diazonium salt, which in turn reacts with N-(1-naphthyl)-ethylenediamine to form a red-violet azo dye that is determined by a reflectometer. The NO_2_ present in the sample is determined by analyzing, without the reduction step.

The chopped samples were ground with pure water using a mortar and pestle. The resulting samples were centrifuged at a speed of 9000 rpm for 10 min at 25 °C using a fast refrigerated centrifuge (Hettich MIKRO 220 R, Kirchlengern, Germany) in Vietnam and a high-speed refrigerated microcentrifuge (TOMY MX-307, Katsushika, Japan) in Japan. The supernatant was used to measure NO_3_-N and NO_2_-N using a reflectometer (RQ-flex Plus 10, Merck Inc., Darmstadt, Germany).

The NO_3_ and NO_2_ were expressed as mg kg^−1^ FW(1)$$Nitrate\;content\left(\frac{mg}{kg\;FW}\right)=measurement\;value\left(\frac{mg}{L}\right)\times \frac{volume\;of\;water\left(mL\right)}{sample\;weight\left(g\right)}\times dillution\;factor$$(2)$$Nitrite\;content\left(\frac{mg}{kg\;FW}\right)=measurement\;value\left(\frac{mg}{L}\right)\times \frac{volume\;of\;water\left(mL\right)}{sample\;weight\left(g\right)}\times dillution\;factor$$

### 4.3. Vitamin C Analysis

The vitamin C (ascorbic acid) contents in the vegetables were determined via a reflectometric method, using the method previously explained [38,39]. Ascorbic acid is reduced via yellow molybdophosphoric acid to phosphor molybdenum blue, which is determined by a reflectometer.

To measure vitamin C, chopped samples were ground with 5% metaphosphoric acid solution using a mortar and pestle. The resulting samples were centrifuged at a speed of 9000 rpm for 10 min at 25 °C using a fast refrigerated centrifuge (Hettich MIKRO 220 R, Kirchlengern, Germany) in Vietnam and a high-speed refrigerated microcentrifuge (TOMY MX-307, Katsushika, Japan) in Japan. Finally, an ascorbic acid strip (Reflectoquant^®^, Merck Inc., Darmstadt, Germany) was immersed in the solution and placed in the reflectometer (RQ-flex Plus 10, Merck Inc., Darmstadt, Germany). The results were expressed in mg 100 g^−1^ FW.(3)$$Vitamin\;C\;content\left(\frac{mg}{100\;g\;FW}\right)=measurement\;value\left(\frac{mg}{L}\right)\times \frac{volume\;of\;acid\;\left(mL\right)}{sample\;weight\left(g\right)\times 10}\times dillution\;factor$$

### 4.4. Risk Assessment

For risk assessment, the NO_3_, NO_2_, and vitamin C content in each type of vegetable and the amount of vegetable consumed were suggested as the basis of the calculation.

NO_3_ is soluble in water, and several reduction factors during the preparation and cooking of food should be taken into account, including a 15% reduction from washing vegetables and a 50% reduction from heat treatment [40]. Additionally, vitamin C, at a dose of 1000 mg, can act as an effective inhibitor of endogenous nitrosation [25].

According to the WHO, a healthy diet includes a daily intake of 400 g of fruits and vegetables [34]; therefore, the total risk for an individual weighing 60 kg consuming 200 g per day of the leafy vegetable samples or 100 g per day of the herb samples was calculated using following equations:(4)$$Nitrate\;intake\left(mgnit\;rate\right)=Nitrate\;content\left(\frac{mg}{100\;g}\right)\times \left(1-0.15-0.5\right)\times vegetable\;number$$(5)$$Nitrate\;risk\left(mgnit\;rite\right)=Nitrate\;intake\times 0.05\times (1-\frac{Vitamin\;C}{1000})$$(6)$$Nitrite\;intake\left(mgnit\;rite\right)=Nitrite\;content\left(\frac{mg}{100\;g}\right)\times \left(1-0.15-0.5\right)\times vegetable\;number$$(7)$$Total\;risk\left(mgnit\;rite\right)=Nitrate\;risk+Nitrite\;risk$$

0.15: nitrate reduction obtained through washing.0.50: nitrate reduction obtained through heat treatment.0.05: conversion rate of nitrate to nitrite.

The vegetable number for 200 g of consumed vegetables is 2; for 100 g of consumed herbs, it is 1.

### 4.5. Statistical Analysis

All experiments were performed in triplicate, and the results were presented as the mean ± SD (standard deviation). Before conducting parametric tests, the data characteristics were tested for homogeneity of variance and normality. The Shapiro–Wilk test was conducted, including a visual inspection of the histograms, and a standard Q–Q plot, which showed that data were normally distributed, as the nun hypothesis was accepted at *p* < 0.05. A two-way analysis of variance was carried out to test for significant differences, and subsequently, a post hoc Tukey test and a paired *t*-test at a significant level of *p* < 0.05 were performed to locate the differences using R (version 4.3.2).

## 5. Conclusions

This study represents the first evaluation of the effects of organic versus synthetic fertilizers on nitrogen accumulation and vitamin C content in leafy vegetables and herbs commonly consumed in South and Southeast Asia. The findings indicate that the type of fertilizer significantly influences both NO_3_ accumulation and vitamin C content in these vegetables. Specifically, organic fertilizers were found to significantly reduce NO_3_ accumulation in amaranth greens, katuk, morning glory, squash leaves, vine spinach, romaine lettuce, komatsuna, and spinach. Additionally, the use of organic fertilizers led to an increase in vitamin C content in amaranth greens, katuk, morning glory, squash leaves, and spinach. In terms of NO_2_ accumulation, the results were generally similar between vegetables grown with organic and synthetic fertilizers, except for katuk and Vietnamese perilla, where differences were observed. While organic fertilizers offer numerous benefits, they also exhibit drawbacks, such as decreased nutrient levels, gradual nutrient release, and inconsistent nutrient availability. If not managed properly, these aspects may lead to reduced crop yields or delayed growth in comparison to those for chemical fertilizers. Thus, applying organic fertilizers should be customized to meet the unique requirements of the crops and the specific conditions of the soil to achieve the best results. These findings underscore the potential benefits of using organic fertilizers to enhance the nutritional quality of leafy vegetables and herbs in this region. Most of the vegetables and herbs in Northern Vietnam, including amaranth greens, katuk, morning glory, squash leaves, vine spinach, Vietnamese basil, and Vietnamese perilla, exhibited low levels of NO_3_ and NO_2_ and high levels of vitamin C. Therefore, they are considered toxicologically safe. However, there are potential health risks associated with the consumption of leaf lettuce, komatsuna, and spinach. Our findings also revealed that NO_3_ accumulates in the midrib and petiole, while vitamin C accumulates in the blade.

### Future Perspectives

Addressing the trade-offs between lower NO_3_ accumulation and potential yield reduction in organically grown vegetables requires an integrated approach. A combination of effective nutrient management, strategic crop selection, and sustainable farming practices can help optimize both yield and nutrient quality. The adoption of precision agriculture, crop rotation, and enhanced organic fertilization methods can contribute to a more balanced and sustainable organic farming system, minimizing environmental impacts while ensuring high-quality vegetable production.

The long-term future research direction highlights the importance of adopting a more unified strategy to comprehend fertilizer effectiveness and its influence on soil quality, plant development, and ecological sustainability. Extended studies and cross-disciplinary investigations, which include soil science, plant biology, and environmental simulation, will be vital for creating sustainable fertilizer management techniques in farming.

## Figures and Tables

**Figure 1 plants-14-00917-f001:**
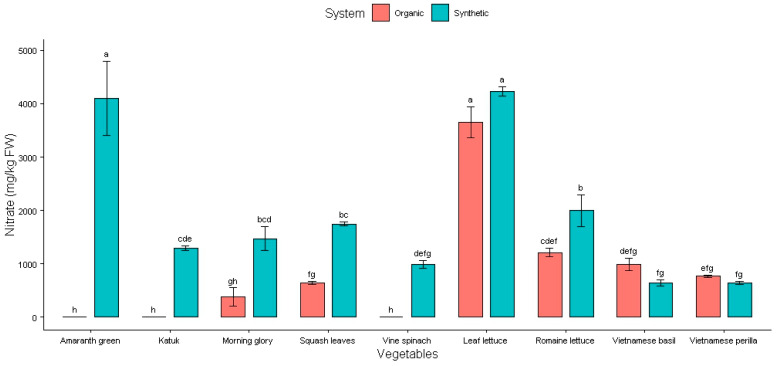
Effect of fertilizers on nitrate content in Vietnamese vegetables. Columns with different letters indicate significant differences via Tukey test (*p* < 0.05).

**Figure 2 plants-14-00917-f002:**
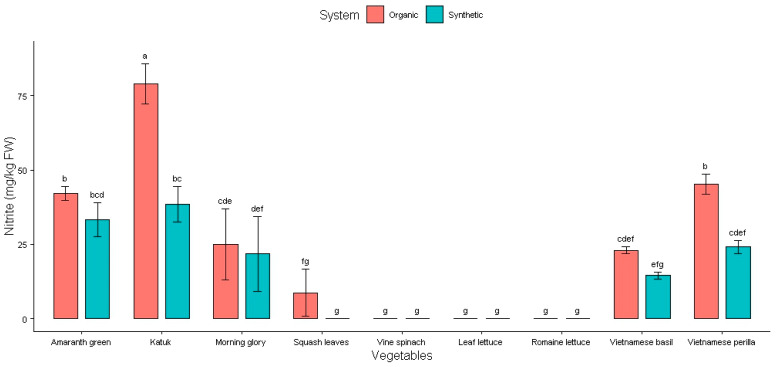
Effect of fertilizers on nitrite content in Vietnamese vegetables. Columns with different letters indicate significant differences via Tukey test (*p* < 0.05).

**Figure 3 plants-14-00917-f003:**
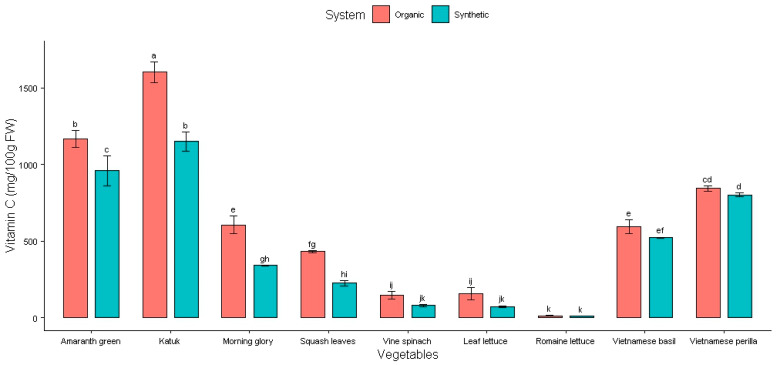
Effect of fertilizers on vitamin C content in Vietnamese vegetables. Columns with different letters indicate significant difference via Tukey test (*p* < 0.05).

**Figure 4 plants-14-00917-f004:**
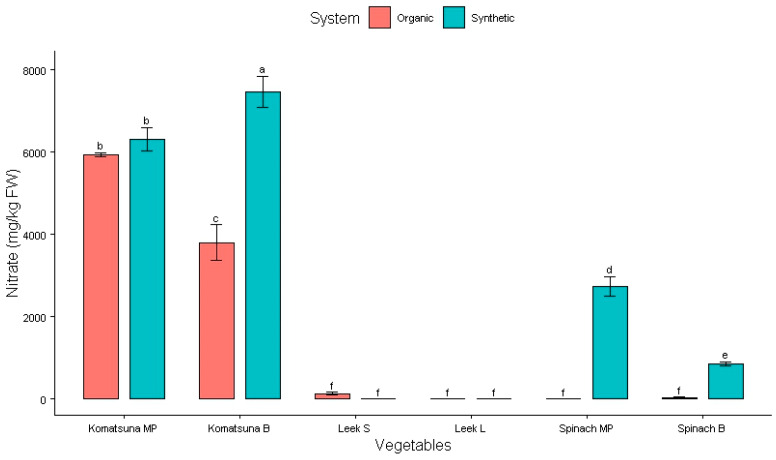
Effect of fertilizers on NO_3_ content in Japanese vegetables. Columns with different letters indicate significant differences via Tukey test (*p* < 0.05). MP: midrib and petiole; B: blade; S: stem; L: leaf.

**Figure 5 plants-14-00917-f005:**
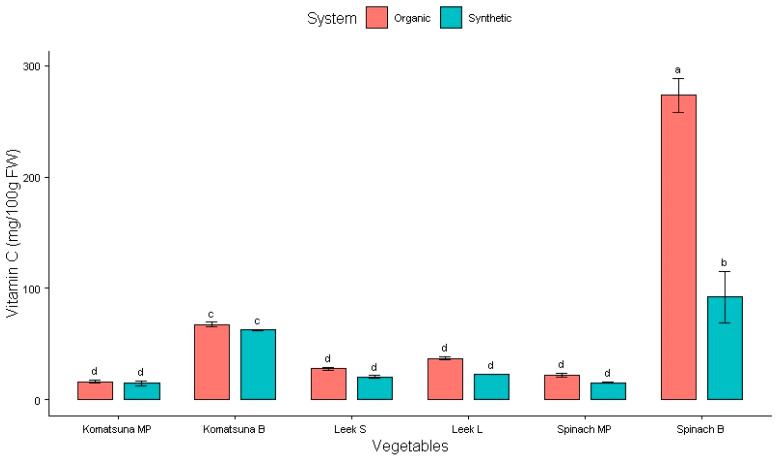
Effect of fertilizers on vitamin C content in Japanese vegetables. Columns with different letters are significantly different (*p* < 0.05) according to the Tukey test. MP: midrib and petiole; B: blade; S: stem; L: leaf.

**Figure 6 plants-14-00917-f006:**
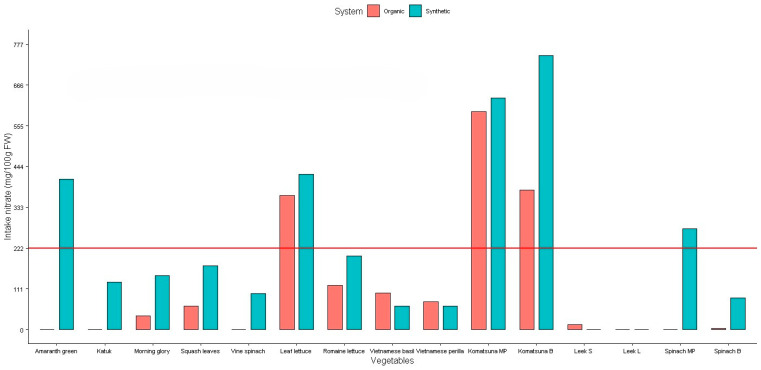
Daily intake of nitrate from studied vegetables. Red line: the acceptable daily intakes of nitrate for a 60 kg adult.

**Figure 7 plants-14-00917-f007:**
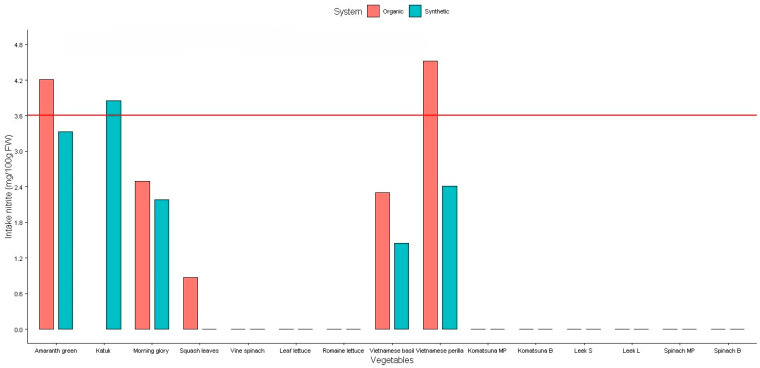
Daily intake of nitrite from studied vegetables. Red line: the acceptable daily intakes of nitrite for a 60 kg adult.

**Figure 8 plants-14-00917-f008:**
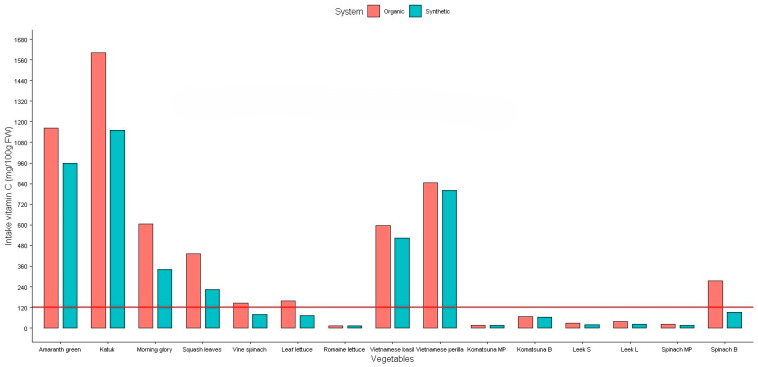
Daily intake of vitamin C from studied vegetables. Red line: the recommended daily intake of vitamin C for breastfeeding women.

**Figure 9 plants-14-00917-f009:**
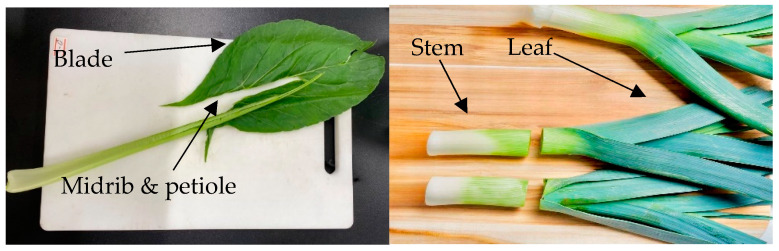
Separated parts of the sample.

**Table 1 plants-14-00917-t001:** Effect of fertilizers on nitrate content in each type of the Vietnamese vegetables.

Vegetables	Fertilizer	Nitrate Content (mg/kg FW)	
Amaranth greens	Organic	0 ± 0	b
	Synthetic	4099 ± 698	a
Katuk	Organic	0 ± 0	b
	Synthetic	1290 ± 44	a
Morning glory	Organic	380 ± 169	b
	Synthetic	1472 ± 223	a
Squash leaves	Organic	643 ± 34	b
	Synthetic	1746 ± 41	a
Vine spinach	Organic	0 ± 0	b
	Synthetic	988 ± 76	a
Leaf lettuce	Organic	3654 ± 292	a
	Synthetic	4232 ± 90	a
Romaine lettuce	Organic	1208 ± 81	b
	Synthetic	2001 ± 296	a
Vietnamese basil	Organic	994 ± 115	b
	Synthetic	640 ± 61	a
Vietnamese perilla	Organic	765 ± 16	b
	Synthetic	640 ± 34	a

The same vegetables with different letters indicate significant differences via paired *t*-test (*p*< 0.05).

**Table 2 plants-14-00917-t002:** Effect of fertilizer on nitrite content in each type of Vietnamese vegetable.

Vegetables	Fertilizer	Nitrite Content (mg/kg FW)	
Amaranth greens	Organic	42 ± 2	a
	Synthetic	33 ± 6	b
Katuk	Organic	79 ± 7	a
	Synthetic	39 ± 6	b
Morning glory	Organic	25 ± 12	a
	Synthetic	22 ± 13	a
Squash leaves	Organic	9 ± 8	a
	Synthetic	0 ± 0	a
Vine spinach	Organic	0 ± 0	a
	Synthetic	0 ± 0	a
Leaf lettuce	Organic	0 ± 0	a
	Synthetic	0 ± 0	a
Romaine lettuce	Organic	0 ± 0	a
	Synthetic	0 ± 0	a
Vietnamese basil	Organic	23 ± 1	a
	Synthetic	14 ± 1	b
Vietnamese perilla	Organic	45 ± 3	a
	Synthetic	24 ± 2	b

The same vegetables with different letters indicate significant differences via paired *t*-test (*p* < 0.05).

**Table 3 plants-14-00917-t003:** Effect of fertilizer on vitamin C content in each type of Vietnamese vegetable.

Vegetables	Fertilizer	Vitamin C (mg/100 g FW)	
Amaranth greens	Organic	1165 ± 55	a
	Synthetic	959 ± 99	b
Katuk	Organic	1601 ± 66	a
	Synthetic	1149 ± 63	b
Morning glory	Organic	605 ± 58	a
	Synthetic	340 ± 0	b
Squash leaves	Organic	432 ± 7	a
	Synthetic	225 ± 17	b
Vine spinach	Organic	147 ± 24	a
	Synthetic	80 ± 7	b
Leaf lettuce	Organic	158 ± 39	a
	Synthetic	73 ± 5	b
Romaine lettuce	Organic	14 ± 1	a
	Synthetic	12 ± 0	b
Vietnamese basil	Organic	595 ± 44	a
	Synthetic	522 ± 4	b
Vietnamese perilla	Organic	844 ± 17	a
	Synthetic	802 ± 11	b

The same vegetables with different letters indicate significant differences via paired *t*-test (*p* < 0.05).

**Table 4 plants-14-00917-t004:** Effect of fertilizers on nitrate content in each type of Japanese vegetable.

Vegetables	Fertilizer	Nitrate Content (mg/kg FW)	
Komatsuna B	Organic	3802 ± 436	b
	Synthetic	7467 ± 387	a
Komatsuna MP	Organic	5934 ± 52	a
	Synthetic	6312 ± 285	a
Leek L	Organic	0 ± 0	a
	Synthetic	0 ± 0	a
Leek S	Organic	133 ± 35	a
	Synthetic	0 ± 0	b
Spinach B	Organic	34 ± 16	b
	Synthetic	864 ± 46	a
Spinach MP	Organic	0 ± 0	b
	Synthetic	2742 ± 237	a

The same vegetables with different letters indicate significant differences via paired *t*-test (*p*< 0.05). MP: midrib and petiole; B: blade; S: stem; L: leaf.

**Table 5 plants-14-00917-t005:** Nitrate content in Japanese vegetables, depending on the part of the vegetable.

Vegetables	Fertilizer	Part	Nitrate Content (mg/kg FW)	
Komatsuna	Organic	Blade	3802 ± 436	a
		Midrib and Petiole	5934 ± 52	b
	Synthetic	Blade	7467 ± 387	a
		Midrib and Petiole	6312 ± 285	b
Leek	Organic	Leaf	0 ± 0	a
		Stem	133 ± 35	b
	Synthetic	Leaf	0 ± 0	a
		Stem	0 ± 0	a
Spinach	Organic	Blade	34 ± 16	a
		Midrib and Petiole	0 ± 0	a
	Synthetic	Blade	864 ± 46	b
		Midrib and Petiole	2742 ± 237	a

The same vegetables with different letters indicate significant differences via paired *t*-test (*p* < 0.05).

**Table 6 plants-14-00917-t006:** Effect of fertilizers on vitamin C content in each type of Japanese vegetable.

Vegetables	Fertilizer	Vitamin C (mg/100 g FW)	
Komatsuna B	Organic	68 ± 2	a
	Synthetic	63 ± 1	a
Komatsuna MP	Organic	16 ± 1	a
	Synthetic	15 ± 2	a
Leek L	Organic	37 ± 1	a
	Synthetic	23 ± 0	b
Leek S	Organic	28 ± 1	a
	Synthetic	21 ± 1	b
Spinach B	Organic	273 ± 15	a
	Synthetic	92 ± 23	b
Spinach MP	Organic	22 ± 2	a
	Synthetic	15 ± 1	b

The same vegetables with different letters indicate significant differences via paired *t*-test (*p* < 0.05). MP: midrib and petiole; B: blade; S: stem; L: leaf.

**Table 7 plants-14-00917-t007:** Vitamin C content in Japanese vegetables, depending on the part of the vegetable.

Vegetables	Fertilizer	Part	Vitamin C Content (mg/100 g FW)	
Komatsuna	Organic	Blade	68 ± 2	a
		Midrib and Petiole	16 ± 1	b
	Synthetic	Blade	63 ± 1	a
		Midrib and Petiole	15 ± 2	b
Leek	Organic	Leaf	37 ± 1	a
		Stem	28 ± 1	b
	Synthetic	Leaf	23 ± 0	a
		Stem	21 ± 1	b
Spinach	Organic	Blade	273 ± 15	a
		Midrib and Petiole	22 ± 2	b
	Synthetic	Blade	92 ± 23	a
		Midrib and Petiole	15 ± 1	b

The same vegetables with different letters indicate significant differences via paired *t*-test (0.05).

**Table 8 plants-14-00917-t008:** Total risk assessment of studied vegetables and herbs.

Vegetables	System	VTMC (mg/100 g FW)	Nitrate (mg/kg FW)	Nitrite (mg/kg FW)	Nitrate Risk (mg NO_2_)	Nitrite Risk (mg NO_2_)	Total Risk (mg NO_2_)
Amaranth greens	Synthetic	959	4099	33	0.6	2.3	2.9
Amaranth greens	Organic	1165	0	42	0.0	2.9	2.9
Katuk	Synthetic	1149	1290	39	0.0	2.7	2.7
Katuk	Organic	1601	0	79	0.0	5.5	**5.5** *
Morning glory	Synthetic	340	1472	22	3.4	1.5	**4.9**
Morning glory	Organic	605	380	25	0.5	1.7	2.3
Squash leaves	Synthetic	225	1746	0	4.7	0.0	**4.7**
Squash leaves	Organic	432	643	9	1.3	0.6	1.9
Vine spinach	Synthetic	80	988	0	3.2	0.0	3.2
Vine spinach	Organic	147	0	0	0.0	0.0	0.0
Leaf lettuce	Synthetic	73	4232	0	13.7	0.0	**13.7**
Leaf lettuce	Organic	158	3654	0	10.8	0.0	**10.8**
Romaine lettuce	Synthetic	12	2001	0	6.9	0.0	**6.9**
Romaine lettuce	Organic	14	1208	0	4.2	0.0	**4.2**
Vietnamese basil	Synthetic	522	640	15	0.5	0.5	1.0
Vietnamese basil	Organic	595	994	23	0.7	0.8	1.5
Vietnamese perilla	Synthetic	802	640	24	0.2	0.8	1.1
Vietnamese perilla	Organic	844	765	45	0.2	1.6	1.8
Komatsuna MP	Synthetic	15	6312	0	21.8	0.0	**21.8**
Komatsuna MP	Organic	16	5934	0	20.4	0.0	**20.4**
Komatsuna B	Synthetic	63	7467	0	24.5	0.0	**24.5**
Komatsuna B	Organic	68	3802	0	12.4	0.0	**12.4**
Leek S	Synthetic	21	0	0	0.0	0.0	0.0
Leek S	Organic	28	133	0	0.5	0.0	0.5
Leek L	Synthetic	23	0	0	0.0	0.0	0.0
Leek L	Organic	37	0	0	0.0	0.0	0.0
Spinach MP	Synthetic	16	2742	0	9.4	0.0	**9.4**
Spinach MP	Organic	22	0	0	0.0	0.0	0.0
Spinach B	Synthetic	92	864	0	2.7	0.0	2.7
Spinach B	Organic	273	34	0	0.1	0.0	3.4

* The bolded numbers indicate the daily intake of nitrite significantly exceeds 3.6 mg.

**Table 9 plants-14-00917-t009:** Samples in the study.

Common and Scientific Name	Sample Location	Number of Samples	Country
Amaranth greens *Amaranthus viridis*	Supermarket	6 (18) *	Vietnam
Katuk *Sauropus androgynus* (L.) Merr.	Supermarket	6 (18)	Vietnam
Squash leaves *Cucurbita moschata* Duchesne	Local farm	6 (18)	Vietnam
Morning glory *Ipomoea aquatica*	Supermarket	6 (18)	Vietnam
Vine spinach *Basella rubra*	Supermarket	6 (18)	Vietnam
Leaf lettuce *Lactuca sativa* var. crispa L.	Supermarket	6 (18)	Vietnam
Romaine lettuce *Lactuca sativa* var. longifolia Lam., var. romana Hort. in Bailey	Supermarket	6 (18)	Vietnam
Vietnamese basil *Ocimum basilicum* L.	Supermarket	6 (18)	Vietnam
Vietnamese perilla *Perilla frutescens* var. *acuta*	Supermarket	6 (18)	Vietnam
Komatsuna *Brassica rapa* var. *perviridis*	Ibaraki Agricultural Institute	6 (30)	Japan
Leek *Allium ampeloprasum* var. *porrum*	Ibaraki Agricultural Institute	6 (30)	Japan
Spinach *Spinacia oleracea*	Ibaraki Agricultural Institute	6 (30)	Japan

*: Numbers in () indicate values of sub-samples.

## Data Availability

Data are contained within the article.

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
