# Peer review of "Effect of Organic and Synthetic Fertilizers on Nitrate, Nitrite, and Vitamin C Levels in Leafy Vegetables and Herbs"

_plants, 2025, doi:10.3390/plants14060917_

Round 1
Reviewer 1 Report
Comments and Suggestions for Authors
The object of the presented article is an evaluation of the effect of two fertilization practices (organic and synthetic) on nutrient and toxic levels in some leafy vegetables. The study is interesting and valuable from both a scientific and practical point of view. Organic crop production (especially in vegetable crops) and the risks related to conventional production are current topics in modern agriculture. In my opinion, the main disadvantage of the study is the fact that most of the studied species (Amaranth green, Katuk, Squash leaves, Water spinach, Vine spinach, Leaf lettuce, Romaine lettuce, Vietnamese basil, Vietnamese perilla) were not grown in scientific experiment conditions but were purchased as random products from supermarkets or farms. It is not known what types of conventional and organic fertilizers were used in the cultivation of the different crops. Probably, these are different fertilizers. To improve the article, I have the following questions and comments:
Materials and methods
Line 368 “*: Number in () indicates values of sub-samples” – Why is the number of subsamples larger (30) in Komatsuna, Leek and Spinach?
Results
Lines 85-86 “The nitrate accumulation in observed vegetables and herbs from Northern Vietnam from July 2023 to August 2024 was presented in Figure 1 and Table 1.” What is the difference between Figure 1 and Table 1 - is it just the statistical processing? If the data are the same, why are the letter designations different?
Line 113 „The nitrite content of these vegetables is presented in Figure 2 and Table 2.“ The same question.
Lines 122-123 „The average levels of nitrites, except amaranth green, katuk, Vietnamese basil and Vietnamese perilla, were very similar (P > 0.05) in both organic and synthetic fertilizer 123 (Tables 2).“ Here, it is important to note that according to the data in Figure 2, there is no significant difference in nitrate content in amaranth green, morning glory, squash leaves, Vietnamese basil.
Lines 106-107 “…………..and organic fertilizer effectively reduced the nitrate content in leafy vegetables, especially in amaranth 107 green.” It cannot be said with certainty whether the organic fertilizer reduces nitrate content in leaves or accumulates in smaller quantities.
Lines 135-137 “The lowest vitamin C value was in Romaine lettuce synthetic (11.82 ± 0.39 mg/ 100g FW) and Romaine lettuce synthetic (13.61 ± 1.34 mg/ 100g FW).” One of the lettuces must be organic.
Line 171 „Table 5. Nitrate content in Japan vegetables depending on part of vegetables.“ How would you explain the high nitrate content in organic Leek L (132.59) compared to conventional Leek L (0.0)?
Discussion
Lines 229-231 “This is the first report that compares the nitrates/nitrites and vitamin C levels of a wide range of South and Southeast Asian leafy vegetables and herbs under the effects of organic versus synthetic fertilizers.” It is too pretentious to say South and Southeast Asia since only two countries are included in the study – Japan and Vietnam.
Author Response
Comment 1: Line 368 “*: Number in () indicates values of sub-samples” – Why is the number of subsamples larger (30) in Komatsuna, Leek, and Spinach?
Response 1: Thank you for your insightful observation. The Ibaraki Agricultural Institute, located in the Kanto area of Japan, supplied komatsuna, leek, and spinach. For each condition, whether organic or conventional, five plants were collected each time, resulting in a greater number of subsamples compared to those collected from Vietnam.
Comment 2: Lines 85-86 “The nitrate accumulation in observed vegetables and herbs from Northern Vietnam from July 2023 to August 2024 was presented in Figure 1 and Table 1.” What is the difference between Figure 1 and Table 1 - is it just the statistical processing? If the data are the same, why are the letter designations different?
Response 2: We greatly appreciate your attention to detail. The data in Figure 1 were analyzed using the Tukey HSD test, which compares various vegetable types across different fertilizer applications. In contrast, Table 1 was analyzed using a paired t-test to examine the impact of fertilizers on the same vegetable types. Although the underlying data remain the same, the statistical methods used differ, leading to distinct letter assignments.
Comment 3: Line 113 “The nitrite content of these vegetables is presented in Figure 2 and Table 2“. The same question.
Response 3: Thank you for raising this point. The explanation provided in the previous response also applies here, as Figure 2 and Table 2 follow the same statistical approach as Figure 1 and Table 1.
Comment 4: Lines 122-123 “The average levels of nitrites, except amaranth green, katuk, Vietnamese basil, and Vietnamese perilla, were very similar (P > 0.05) in both organic and synthetic fertilizers (Table 2).” Here, it is important to note that according to the data in Figure 2, there is no significant difference in nitrate content in amaranth green, morning glory, squash leaves, and Vietnamese basil.
Response 4: Thank you for your keen observation. In Figure 2, the Tukey test was used to analyze 18 different vegetable samples to determine which had the highest levels of nitrogen compounds under different fertilizer conditions. The results showed that katuk, when grown with organic fertilizer, had the highest nitrite accumulation among leafy vegetables. There were no significant differences in nitrite levels among katuk (with synthetic fertilizer), amaranth green (with both organic and synthetic fertilizers), morning glory (with both fertilizers), and Vietnamese basil (with both fertilizers). Similarly, squash leaves showed no significant differences compared to vine spinach, leaf lettuce, and Romaine lettuce, all of which had undetectable nitrite levels under both fertilizer conditions. However, when a paired t-test was applied to assess the effect of fertilizers on specific vegetables and herbs, significant differences in nitrite levels were observed in amaranth green, katuk, Vietnamese basil, and Vietnamese perilla.
Comment 5: Lines 106-107 “…………and organic fertilizer effectively reduced the nitrate content in leafy vegetables, especially in amaranth green.” It cannot be said with certainty whether the organic fertilizer reduces nitrate content in leaves or accumulates in smaller quantities.
Response 5: Thank you for pointing this out. We fully agree with your suggestion and have revised the statement accordingly. This correction can be found in the revised manuscript on page 3, line 108, and page 7, lines 161.
Comment 6: Lines 135-137 “The lowest vitamin C value was in Romaine lettuce synthetic (11.82 ± 0.39 mg/100g FW) and Romaine lettuce synthetic (13.61 ± 1.34 mg/100g FW).” One of the lettuces must be organic.
Response 6: We appreciate your careful review. You are absolutely right, and we have corrected this mistake in the revised manuscript. The change can be found on page 5, line 138.
Comment 7: Line 171 “Table 5. Nitrate content in Japan vegetables depending on part of vegetables.” How would you explain the high nitrate content in organic Leek L (132.59) compared to conventional Leek L (0.0)?
Response 7: Thank you for this important question. A paired t-test revealed significant differences between organic and synthetic fertilizers for komatsuna leaves, leek stalks, spinach leaves, and the midrib and petiole of spinach. Interestingly, organic leek exhibited higher nitrate content than conventional leek. The exact reason for this discrepancy remains unclear, as we lacked detailed information about the fertilizers used for further evaluation.
Comment 8: Lines 229-231 “This is the first report that compares the nitrates/nitrites and vitamin C levels of a wide range of South and Southeast Asian leafy vegetables and herbs under the effects of organic versus synthetic fertilizers.” It is too pretentious to say South and Southeast Asia since only two countries are included in the study – Japan and Vietnam.
Response 8: Thank you for highlighting this issue. We agree and have revised the statement to reflect the study’s scope more accurately. This correction can be found on page 11, line 233.

Reviewer 2 Report
Comments and Suggestions for Authors
Scientific Review of the Article: "Effect of Organic and Synthetic Fertilizers on Nutrient Composition and Toxin Levels in Leafy Vegetables and Herbs"
Introduction The introduction of the article provides a strong foundation for the study, emphasizing the importance of understanding how fertilizer type influences nutrient composition and toxin accumulation in leafy vegetables. The authors highlight key concerns, including nitrate and nitrite accumulation, as well as vitamin C content, which are directly related to human health and environmental sustainability. The cited literature is relevant and reflects the current state of research in the field. However, expanding the literature review to include more comparative studies on organic versus synthetic fertilizers would further enhance the justification of the study's originality.
Research Design and Methodology The research design is well-structured, employing a comparative approach to assess the effects of organic and synthetic fertilizers on different leafy vegetables and herbs. The selection of analytical methods, including nitrate and nitrite quantification, vitamin C analysis, and statistical evaluation, is appropriate and ensures robust data interpretation. The inclusion of vegetables from both Vietnam and Japan adds an interesting comparative dimension to the study.
However, the methodology section could benefit from additional details on the sampling strategy, including the number of replicates per condition and the specific environmental factors that may have influenced the results. Moreover, providing a discussion on the potential variability in nutrient absorption among different plant species would strengthen the study's conclusions.
Presentation of Results The results are presented clearly, with well-structured tables and figures that facilitate data interpretation. The observed differences in nitrate accumulation between organically and synthetically fertilized vegetables are well-documented, with notable findings such as lower nitrate levels and higher vitamin C content in organically grown produce. The statistical analysis supports the validity of these findings.
A more detailed discussion on the potential biochemical mechanisms underlying these differences would add value to the interpretation. Additionally, discussing the implications of elevated nitrate levels in certain vegetables, particularly those not currently regulated, would enhance the study’s impact.
Conclusions and Summary The conclusions are well-aligned with the presented results and highlight the potential benefits of organic fertilizers in reducing nitrate accumulation while enhancing vitamin C content. The study successfully underscores the importance of organic vegetable cultivation in promoting human health and environmental sustainability.
However, a more extensive discussion on the potential limitations of organic fertilizers, such as lower yield or nutrient variability, would provide a more balanced perspective. Future research directions, including long-term studies on soil health and fertilizer efficiency, should also be suggested.
Evaluation of English Language The English language in the article is generally clear and well-structured. The scientific terminology is appropriately used, and the manuscript is comprehensible. However, minor refinements in phrasing and sentence structure could improve readability and flow.
Questions for the Authors:
-
Could you provide more details on the potential environmental factors that may have influenced nitrate and vitamin C accumulation in the studied vegetables?
-
Have you considered assessing additional plant metabolites that may also be influenced by fertilizer type?
-
How do you propose addressing the potential trade-offs between lower nitrate accumulation and possible yield reduction in organically grown vegetables?
Author Response
Comments 1: Introduction The introduction of the article provides a strong foundation for the study, emphasizing the importance of understanding how fertilizer type influences nutrient composition and toxin accumulation in leafy vegetables. The authors highlight key concerns, including nitrate and nitrite accumulation, as well as vitamin C content, which are directly related to human health and environmental sustainability. The cited literature is relevant and reflects the current state of research in the field. However, expanding the literature review to include more comparative studies on organic versus synthetic fertilizers would further enhance the justification of the study's originality.
Response: Thank you for your insightful comment. We appreciate your suggestion to expand the literature review with more comparative studies on organic versus synthetic fertilizers. In response, we have incorporated additional references that further support the originality of our study. These revisions can be found in the revised manuscript on page 2, lines 54, 57–60.
Comments 2: Research Design and Methodology The research design is well-structured, employing a comparative approach to assess the effects of organic and synthetic fertilizers on different leafy vegetables and herbs. The selection of analytical methods, including nitrate and nitrite quantification, vitamin C analysis, and statistical evaluation, is appropriate and ensures robust data interpretation. The inclusion of vegetables from both Vietnam and Japan adds an interesting comparative dimension to the study.
However, the methodology section could benefit from additional details on the sampling strategy, including the number of replicates per condition and the specific environmental factors that may have influenced the results. Moreover, providing a discussion on the potential variability in nutrient absorption among different plant species would strengthen the study's conclusions.
Response: We appreciate your valuable feedback on improving the methodology section. In response, we have added details regarding the sampling strategy, including the number of replicates per condition and the specific environmental factors that may have influenced the results. These updates are reflected on page 17, lines 404–415 and 418–424.
Additionally, our research encompassed twelve distinct plant varieties commonly cultivated in Japan and Vietnam. Many of these species have not been extensively studied, resulting in limited data on potential differences in nutrient uptake across various plant types in this specific context.
Comments 3: Presentation of Results The results are presented clearly, with well-structured tables and figures that facilitate data interpretation. The observed differences in nitrate accumulation between organically and synthetically fertilized vegetables are well-documented, with notable findings such as lower nitrate levels and higher vitamin C content in organically grown produce. The statistical analysis supports the validity of these findings.
A more detailed discussion on the potential biochemical mechanisms underlying these differences would add value to the interpretation. Additionally, discussing the implications of elevated nitrate levels in certain vegetables, particularly those not currently regulated, would enhance the study’s impact.
Response: Thank you for your thoughtful comment. We acknowledge the importance of discussing the biochemical mechanisms behind the observed differences in nitrate accumulation and vitamin C content. In response, we have expanded this discussion in the revised manuscript on page 13, lines 297–313.
Furthermore, we have also addressed the implications of elevated nitrate levels in certain vegetables, particularly those that are not currently regulated. This discussion can be found on page 12, lines 244–257, page number 13, line 321–328
Comments 4: Conclusions and Summary The conclusions are well-aligned with the presented results and highlight the potential benefits of organic fertilizers in reducing nitrate accumulation while enhancing vitamin C content. The study successfully underscores the importance of organic vegetable cultivation in promoting human health and environmental sustainability.
However, a more extensive discussion on the potential limitations of organic fertilizers, such as lower yield or nutrient variability, would provide a more balanced perspective. Future research directions, including long-term studies on soil health and fertilizer efficiency, should also be suggested.
Response: We appreciate your suggestion to provide a more balanced perspective on the potential limitations of organic fertilizers. In response, we have expanded our discussion on aspects such as lower yield and nutrient variability, which can be found on page 19, lines 501–506.
Additionally, we have incorporated future research directions, including long-term studies on soil health and fertilizer efficiency, to further enhance the study’s impact. These additions are located on page 19, lines 531–542.
Comments 5: Evaluation of English Language the English language in the article is generally clear and well-structured. The scientific terminology is appropriately used, and the manuscript is comprehensible. However, minor refinements in phrasing and sentence structure could improve readability and flow.
Response: Thank you for your feedback on the manuscript’s language clarity. We have carefully revised the text to refine phrasing and sentence structure, improving readability and flow while maintaining scientific accuracy.
Comments 6: Questions for the Authors:
- Could you provide more details on the potential environmental factors that may have influenced nitrate and vitamin C accumulation in the studied vegetables?
- Have you considered assessing additional plant metabolites that may also be influenced by fertilizer type?
- How do you propose addressing the potential trade-offs between lower nitrate accumulation and possible yield reduction in organically grown vegetables?
Response:
- Thank you for your question. We have provided additional details on the environmental factors influencing nitrate and vitamin C accumulation in the studied vegetables. These details can be found on pages 16 and 17, lines 384–395 and 398–405.
- Yes, we have considered assessing additional plant metabolites influenced by fertilizer type. This aspect will be explored further in the next phase of our research, where we plan to analyze the metabolomic profile of the studied crops.
- Addressing the trade-offs between lower nitrate accumulation and potential yield reduction in organically grown vegetables requires an integrated approach. A combination of effective nutrient management, strategic crop selection, and sustainable farming practices can help optimize both yield and nutrient quality. The adoption of precision agriculture, crop rotation, and enhanced organic fertilization methods can contribute to a more balanced and sustainable organic farming system, minimizing environmental impacts while ensuring high-quality vegetable production.
